# Sonographic Evaluation of Medial Iliac Lymph Nodes-to-Aorta Ratio in Dogs

**DOI:** 10.3390/vetsci7010022

**Published:** 2020-02-11

**Authors:** Simonetta Citi, Martina Oranges, Elena Arrighi, Valentina Meucci, Daniele Della Santa, Mannucci Tommaso

**Affiliations:** 1Department of Veterinary Sciences, School of Veterinary Medicine, University of Pisa, 56100 Pisa, Italy; martina.oranges@gmail.com (M.O.); elena.arrighi@hotmail.com (E.A.); valentina.meucci@unipi.it (V.M.); tommy.mannucci@gmail.com (M.T.); 2Vet Hospital H24, 50100 Firenze, Italy; danieledellasanta@yahoo.it

**Keywords:** ultrasonography, dog, medial iliac lymph node, aorta, ratio

## Abstract

Medial iliac lymph nodes drain many districts and are easy to identify during an ultrasound examination of the abdomen. Since there are no reference values for their size in healthy dogs, the aim of this work was to evaluate the size of the medial iliac lymph nodes by using a ratio with the aortic diameter and find a reference range. The population was divided into group A (healthy dogs) and group B, with diseases of the medial iliac lymph nodes. The ratio of length, height and thickness of the medial iliac lymph nodes with the diameter of the aorta were calculated and underwent statistical analysis, *p* < 0.05 was considered statistically significant. Sixty-three patients were enrolled in group A, and 37 in group B. Significant differences were found between the ratio of sick and healthy patients and neoplastic and healthy patients. No significant difference was found between healthy and inflammatory patients. The best cut-off value to discriminate sick and healthy patients was 0.57, with a sensitivity of 78% and a specificity of 71%. The cut-off value of neoplastic and healthy patients was 0.69, with a sensitivity of 89.47% and a specificity of 84.13%. This value is highly predictive of neoplasia.

## 1. Introduction

The medial iliac lymph nodes belong to the iliosacral lymph center and are the largest lymph nodes in this group [1]. They are located laterally to the aorta near its trifurcation, between the deep circumflex iliac and the external iliac arteries [1,2]. They are usually single but can also be double on the left, right, or on both sides [1]. These nodes drain the skin of the dorsal abdominal wall caudal to the last rib, the skin of the pelvis, thigh and stifle, the tail root, the abdominal, pelvic, and lumbar muscles, the muscles and the bones of the pelvic limb, testis, epididymis, spermatic cord, vaginal tunic, cremaster muscle, prostate gland, vagina, vulva, colon, rectum, anus, ureter, bladder, urethra, aorta, spinal cord meninges, inguinal area, and left colic, sacral, and internal iliac lymph nodes [1].

The ultrasonography showed good results in the detection of both peripheral lymph nodes and deeper lymph nodes which are not identifiable by palpation [3,4,5,6,7,8,9]. The medial iliac lymph nodes are anatomically located ventral to L6 and L7. If greatly increased in size, the descending colon and rectum are displaced ventrally and sublumbar lymphadenomegaly can be radiographically diagnosed [10].

Ultrasonographic examination of the medial iliac lymph nodes is currently performed during each abdominal ultrasound scan [3,11].

These lymph nodes are easily identified because they have a fixed landmark and they are not usually hidden by ultrasound artifacts, typical of the gastro-intestinal tract [2,5]. In addition, they are often hypoechoic to adjacent tissues, which helps their ultrasonographic identification [5,12].

In 2010, Mayer detected the medial iliac lymph nodes in 100% of 50 healthy dogs [4].

As these lymph nodes are easily identified, it is important to perform this ultrasonographic evaluation during all ultrasound abdominal examinations. This enables clinicians to diagnose systemic diseases and abnormalities in the various anatomical districts drained by these lymph nodes. 

It is difficult to differentiate between benign and malignant lymphadenopathy because the analyzed features used as distinctive criteria, such as size, echotexture, and shape, may be similar, especially in the early stages of disease [12,13,14].

Several papers describe ultrasonographic features of neoplastic lymph nodes, detected by means of B-Mode ultrasonography and Color Doppler [5,14,15,16]. The short-axis/long-axis ratio seems to be a good indicator for predicting malignancy [8]; however, it does not provide a reliable discrimination because of the overlap between malignant and benign lymph nodes, thus making this approach difficult [3,5,8,17].

Size determination is commonly used for the initial diagnosis and for detecting changes during the follow-up period [5,6,17]. Currently, in human medicine, size is the most widely used criterion to diagnose cervical node metastasis [18]. 

Based on our clinical experience, in some cases, a subjective enlargement of the medial iliac lymph nodes is detected, without necessarily more striking modifications. This may thus raise the question of the level of involvement of lymph nodes, an important factor for both the internist and oncologist.

The great variability of dog breeds has prevented clinicians from establishing a reference range of medial iliac lymph nodes’ sizes. Veterinary studies and anatomy books report different values, and ultrasound evaluations are usually based on a subjective assessment. Mayer reported that ultrasound values of medial iliac lymph node dimensions were lower than the values reported in various anatomy books [4]. Mayer and colleagues are the only authors who studied the correlation between body weight and the right and left medial iliac lymph nodes’ size in dogs [4]. The mean values for small (<16.5 kg), medium (16.5–29.8 kg) and large (>29.8 kg) dogs were respectively, for the right and left medial iliac lymph node, 0.43–0.45 cm, 0.63–0.64 cm, and 0.70–0.75 cm [4].

Considering that there is no standardized measurement related to the dog size, the ultimate objective was to find a reference range in healthy dogs normalized by the size. A second objective was to investigate the change in size of the medial iliac lymph nodes in neoplastic and inflammatory diseases.

## 2. Materials and Methods

Clinical patients seen at the Veterinary Teaching Hospital of the Department of Veterinary Science (University of Pisa, Italy) between January 2018 and December 2018 were separated into two groups. 

GROUP A: dogs with no clinical findings related to systemic diseases or to the anatomical structures drained by the medial iliac lymph nodes (control group).

GROUP B: dogs with systemic/neoplastic diseases (non-treated lymphoma) or with infectious/neoplastic diseases involving anatomical regions drained by the medial iliac lymph nodes.

Both owners’ written consent and approval from the University (Pisa Veterinary School, Italy) Ethical Committee were obtained prior to beginning the study (Pr.N. 0010138/2018). 

All patients underwent a complete abdominal ultrasound examination, which included examination of medial iliac lymph nodes’ evaluation. 

Patients in group A were examined for clinical work-up for surgical neutering or orthopedic radiographic studies. All owners gave their authorization to conduct an abdominal ultrasound, which included left and right medial iliac lymph nodes. The dogs were included in this group after checking that they were in good health through a medical history, laboratory tests, abdominal ultrasound, and phone call follow-up after three months. 

Patients in group B were examined due to systemic diseases or disorders of the anatomical regions drained by medial iliac lymph nodes. In these cases, further examinations were necessary, including an abdominal ultrasound. The inclusion criteria were a clinical examination, laboratory tests, and cytological examination of the affected lymph node. The exclusion criteria were uncooperative dogs, and dogs where it was not possible to identify lymph nodes bilaterally. All patients were released from the Veterinary Teaching Hospital after establishing a definitive diagnosis and appropriate treatment.

Dogs were sedated only if necessary for clinical staging. The protocol for sedation was chosen according to the clinical condition of the patient. 

The ultrasound examinations were performed using an Aplio 400 (Toshiba, Milan, Italy) with either a 12 MHz Linear or a 7.5 MHz micro-convex transducer.

The ultrasound examinations were performed in a quiet environment with low-light and controlled temperature. Two operators gently manually restrained the dogs if they were awake.

After a complete scan of the abdomen, all patients were scanned first in right lateral recumbency to analyze the left medial iliac lymph node and aorta, and then in left lateral recumbency to analyze the right medial iliac lymph node.

Two radiologists with decades of experience (S.C. and T.M.) acquired the images and video files of the medial iliac lymph nodes and aorta, assessing longitudinal and transverse scans. 

In all medial iliac lymph nodes in group B, during the ultrasound examination, ultrasound-guided fine needle aspirate was performed, and these cytological samples were subsequently evaluated by expert cytologists.

Images and video files were stored in a database and all sonographic measurements were performed by a third expert radiologist (D.D.S.), unaware of the patient’s medical history. 

Length, defined as maximal craniocaudal extension (L), and height, defined as maximal dorsoventral extension (H), of the medial iliac lymph nodes were assessed in a longitudinal scan, and thickness (T) was assessed in a transverse scan, at the point of greatest lateromedial amplitude. We also measured the aortic diameter during systole, immediately cranial to its trifurcation, in both scans. Measurements were taken in centimeters, using electronic calipers.

All measurements were entered into an Excel spreadsheet that calculated the ratio of the length, height, and thickness of the medial iliac lymph nodes with the diameter of the aorta in transverse (TAoD) and longitudinal (LAoD) scanning.

Data analysis was performed using Graph Pad Prism software version 7. The Kolmogorov–Smirnov test was performed to evaluate data distribution. Student’s t-test was used to evaluate potential differences between the aortic diameter in the longitudinal scan and the aortic diameter in the transverse scan. The paired-sample Wilcoxon test was used to evaluate any differences between measurements of length, height, and thickness of medial iliac lymph nodes and the same measurements normalized by aortic diameter (in longitudinal scan), in both recumbencies. Spearman rank correlation coefficients and linear regression models were used to assess the correlation between the aortic diameter (in longitudinal scan) and all measurements of both right and left medial iliac lymph nodes, as well as their ratio. Spearman rank correlation coefficients and linear regression models were used to assess the correlation between the weight of the dogs and the height of both right and left medial iliac lymph nodes, as well as their ratios. The analysis was performed on the entire sample but also stratified by the patients’ health status (sick/healthy). Sick patients were also divided in neoplastic and inflammatory groups and compared with the healthy group. The Mann–Whitney test was used to evaluate differences between the ratio of males and females and the ratio of sick and healthy patients. Receiver operating characteristic (ROC) curve analysis was used to identify the optimal diagnostic ratio cut-off in the detection of sick patients, and the optimal combination of sensitivity and specificity was determined. A *p*-value < 0.05 was considered statistically significant. 

## 3. Results

One hundred patients were enrolled in this study, 63 in the control group (group A), and 37 in the group with systemic diseases or disorders of the anatomical regions related to the medial iliac lymph nodes (group B). The median age was 5 years (ranging between 5 months and 16 years), and 53% were female. Multiple breeds and mixed breeds were included in the study and the weights ranged between 5.3 kg and 41 kg (median 21 kg).

In group A, there were 25 males and 38 females, the mean age was 4.5 ± 3.9 years, and the mean weight was 22.5 ± 11.0 kg. Twenty-four dogs were mixed breed while the others belonged to 15 different breeds: Labrador Retriever (11), German Shepherd (5), Australian Shepherd (2), Kurzhaar (2), Jack Russell Terrier (3), Springer Spaniel (4), Maremmano (1), Pitbull (1), Dachshund (2), Chihuahua (2), Setter (2), Boxer (1), Beagle (1), Doberman (1), and Golden Retriever (1).

In group B, there were 22 males and 15 females, the mean age was 7.5 ± 3.8 years, and the mean weight was 21.2 ± 10.10 kg. Fifteen dogs were mixed breed while the others belong to 13 different breeds: Labrador Retriever (1), German Shepherd (2), Kurzhaar (2), Jack Russell Terrier (1), Springer Spaniel (7), Dachshund (1), Chihuahua (1), English Setter (1), Boxer (1), Beagle (1), Doberman (1), Rottweiler (2), and Golden Retriever (1).

The following diseases were diagnosed in patients in group B:16 inflammations caused by ulcerative colitis (4 patients), hemorrhagic cystitis (2 patients), acute prostatitis (3 patients), perianal abscess (2 patients), prostatic abscess (3 patients), stump pyometra (1 patient), and discospondylitis (1 patient)21 neoplasia, of which there were 11 untreated lymphomas and 10 metastases (Table 1).

Measurements of the right and left medial iliac lymph nodes were performed separately. Overall, 63 right lymph nodes and 63 left lymph nodes were evaluated in group A. For group B, only altered lymph nodes were included in the analysis, with a total of 34 left lymph nodes and 35 right lymph nodes evaluated.

The length, height, and thickness of the medial iliac lymph nodes and aortic diameter in longitudinal and transversal scans were assessed. Six ratios were obtained for each lymph node included in this study. 

The paired t-test, in all enrolled dogs, did not show any significant difference (*p* > 0.05) between the aortic diameter in the longitudinal scan and in the transverse scan. Therefore, based on previous evidence [19] which suggests that the aortic diameter in the longitudinal scan was assessed better than in the transversal scan, we decided to use the longitudinal diameter.

The results of the paired-sample Wilcoxon test between measurements of length, height, and thickness of medial iliac lymph nodes and the same measurements normalized by the aortic diameter are presented in Table 2.

The results did not show any significant difference with regards to height, and the ratio of the aortic diameter and height of the lymph node, while significant differences were found for the length, thickness, the ratio of aortic diameter and lymph node length, and the ratio of aortic diameter and lymph node thickness. In all cases, the measurements related to the right lymph nodes were greater.

Spearman’s rank correlation coefficients and linear regression models showed a statistically significant linear positive correlation between aortic diameter (in the longitudinal scan) and all measurements of both the right and left medial iliac lymph nodes (*p* < 0.0001). The lymph node measurements increased with the increase in the aortic diameter. 

Spearman’s rank correlation coefficients and linear regression models did not show a correlation between the body weights of dogs and the ratio between height and the aorta’s diameter of both the right and left medial iliac lymph nodes (*p* > 0.05). 

The Mann–Whitney test did not show any significant difference between the ratio of males and females (*p* > 0.05) and the ratio of young (<18 months) and adult (>18 months) patients (*p* > 0.05). On the other hand, a statistically significant difference was found for the ratio of sick and healthy patients (*p* < 0.0001). 

The Mann–Whitney test did not show any significant difference between the ratio of healthy and inflammatory patients (*p* > 0.05), while a statistically significant difference was found for the ratio of neoplastic and healthy patients (*p* < 0.0001) (Figure 1).

The best cut-off value of the ratio between the aortic diameter and lymph node height to discriminate between sick and healthy patients is reported in Table 3.

## 4. Discussions

Reproducibility is fundamental for any measurement method. In other studies, the ratio with LAoD has been shown to be an effective and reliable method for assessing the size of some anatomical structures. It has been used to quantify the magnification of the left atrium [20], to evaluate the diameter of the portal vein in porto systemic shunts [21], to assess the state of the volume depletion [22], and also to define a range for the renal size in dogs [19].

We believe that this is the first prospective study using the ratio with the aorta to establish a reference value for the size of the medial iliac lymph nodes in dogs and to identify benign or malignant alterations.

To calculate the ratio, the height of the lymph node was considered because it was the only measurement that did not show statistically significant differences between the right and left lymph nodes in healthy subjects. In addition, the height of the lymph node and the LAoD can be measured using the same longitudinal scan, which is easier to perform than the transversal scan [19]. The asymmetries of the length and thickness measurements, also found in other studies, may be due to biological individual variations, errors due to patient movements, or errors due to a different orientation of the probe during the measurements [23]. The diameter of the aorta and the measurements of the lymph nodes (length, height, and thickness) showed a linear correlation. In fact, with an increase in the aorta’s diameter, the values of the three measurements also increased. The same correlation exists between the aorta’s diameter and body weight. Therefore, a larger subject has a larger aorta and medial iliac lymph nodes. A correlation between body weight and the ratio between height and the aorta’s diameter does not exist. The ratio between the medial iliac lymph node measurements and the aorta can thus be used to assess the size of the lymph nodes in all breeds, because this measurement is independent from body weight.

This study showed that the relationship between the height of the medial iliac lymph node and the LaoD is not influenced by the patient’s gender and age. Krol’s study in 2012 showed how the medial iliac lymph nodes, unlike the other lymph nodes, have similar dimensions between young and adult subjects [24]. These results therefore make this ratio usable in clinical practice.

This study also showed that there is a statistically significant difference between the ratios of healthy and pathologic subjects.

The ROC curve analysis established a cut-off for the differentiation between a normal and a pathological lymph node. Comparing the ratios of subjects with pathologies affecting the medial lymph nodes with those of healthy subjects, the cut-off appears to be 0.57, with a sensitivity of 78% and a specificity of 71%. A ROC curve analysis was also performed by separating the ratio of inflammatory to neoplastic lymph nodes.

This highlighted how there is no statistically significant difference between the ratio of healthy compared to inflammatory lymph nodes. In this case, the ROC curve established a cut-off of 0.56 with reduced sensitivity and specificity, at 62.5% and 69.84%, respectively. The accuracy of the ROC curve calculated by evaluating only the neoplastic lymph nodes compared to healthy ones, instead, increased considerably: the sensitivity was 89.47% and the specificity was 84.13%, with a cut-off value of 0.69.

It is not possible to ultrasonographically differentiate lymphadenopathies of an inflammatory or neoplastic origin, because the findings used as distinctive criteria can be similar, especially in the early stages of the disease [12,13,14].

Studies reported in the human and veterinary literature, which evaluate the echographic differences of lymph nodes in benign and malignant pathologies, conclude that it is necessary to evaluate several characteristics and not just one individual parameter [3,5,8,18]. The results of our study show that the dimension alone is very likely to identify the neoplastic character of an iliac lymph node, differentiating it from a normal lymph node, but also from a lymph node that responds to an inflammatory stimulus.

However, a 0.69 cut-off value is not reliable in differentiating a healthy lymph node from one with a current inflammatory pathology.

This study has several limitations. As in other studies that have used this ratio, the level of experience required to obtain reliable ultrasound measurements was not evaluated, however all sonographers with a basic training should be able to perform this type of measurement. Although different effects that may vary its size can also affect the diameter of the aorta, we did not evaluate systemic arterial pressure in group A subjects, nor the degree of hydration. To obviate changes linked to the cardiac cycle, measurements of the aortic diameter were taken from videos when there was maximal luminal measurement. The medial iliac lymph nodes of control dogs were not sampled: although unlikely, neither neoplastic nor inflammatory lymphadenopathy can be completely ruled out. Furthermore, the final diagnosis was based on cytologic examination of nodal aspirates whose overall diagnostic accuracy is lower than histopathology [25]. Finally, the number of dogs with inflammatory diseases was limited. It would have been interesting to also include the class of infectious diseases (leishmania, hemoparasitosis), however for ethical reasons, it was not possible to perform needle aspiration in medial lymph nodes, instead of superficial lymph nodes.

In conclusion, the relationship between the height of the medial iliac lymph nodes and the LAoD can be reliably used to highlight the enlargement of the medial iliac lymph node.

In the early stages of lymphadenopathy, an index that allows the sonographer to reliably detect mild nodal enlargement is a clinically useful tool, and this is especially true when other anomalous morphological findings (shape, edges, echogenicity, ecotype, echogenicity of perinodal fats) have not been observed.

As already demonstrated in numerous studies, a single alteration of ultrasound does not identify the type of pathology present; therefore, evaluation of nodal size alone might be misleading and multiple sonographic features must be assessed in order to properly classify the lymph node.

We have shown that quantitative ecographic evaluations, which take into account the height of the lymph nodes, are highly predictive of the presence of primary or secondary neoplasia of the medial iliac lymph nodes.

## Figures and Tables

**Figure 1 vetsci-07-00022-f001:**
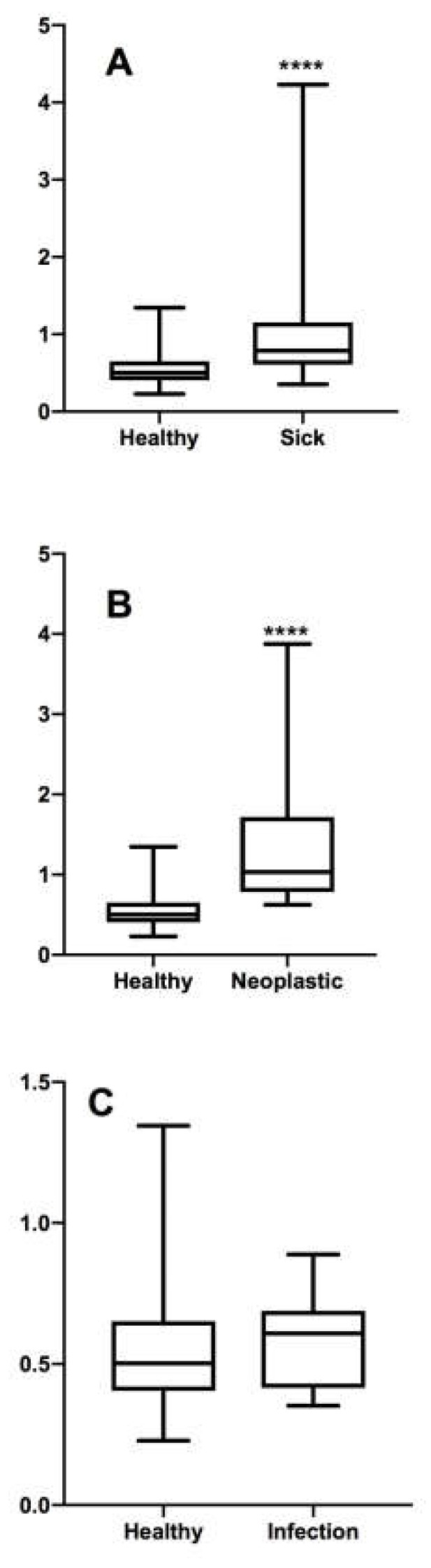
Results of the Mann–Whitney test between (**A**) the ratio of the mean height of right and left medial iliac lymph nodes and the aortic diameter of healthy and sick patients, (**B**) the ratio of the mean height of right and left medial iliac lymph nodes and the aortic diameter of healthy and neoplastic patients, (**C**) the ratio of the mean height of right and left medial iliac lymph nodes and the aortic diameter of healthy and inflammatory patients. **** There are significant difference between two groups.

**Table 1 vetsci-07-00022-t001:** Primary tumors in group B patients with lymph node metastasis.

Metastasis	Primary Tumor	Monolateral	Bilateral
1.	Adenocarcinoma of right caudal abdominal mammary gland	✗	
2.	Mastocytoma hindquarters, left	✗	
3.	Adenocarcinoma of bilateral caudal abdominal mammary gland		✗
4.	Prostatic adenocarcinoma		✗
5.	Penis hemangiosarcoma		✗
6.	Sarcoma left hind limb	✗	
7.	Adenocarcinoma of left caudal abdominal mammary gland	✗	
8.	Bilateral perianal adenocarcinoma		✗
9.	Urethral carcinoma		✗
10.	Adenocarcinoma of right caudal abdominal mammary gland	✗	

**Table 2 vetsci-07-00022-t002:** Results of the paired-sample Wilcoxon test. HR and HL, height of right and left medial iliac lymph nodes. LR and LL, length of right and left medial iliac lymph nodes. TR and TL, thickness of right and left medial iliac lymph nodes. Ao, aortic diameter (in longitudinal scan).

	*p* Value
HR versus HL	*p* > 0.05
HR/Ao versus HL/Ao	*p* > 0.05
LR versus LL	*p* = 0.0016
LR/Ao versus LL/Ao	*p* = 0.0017
TR versus TL	*p* < 0.0001
TR/Ao versus TL/Ao	*p* < 0.001

**Table 3 vetsci-07-00022-t003:** Sensitivity and specificity of different cut-off points of the ratio between the aortic diameter and height of the lymph node to discriminate between sick and healthy patients.

	AUC	95% CI	*p*	Cut-Off	Se (%)	Sp (%)
Sick/healthy	0.79	0.70–0.89	<0.0001	>0.58	78.38 (61.79–90.17)	71.43 (58.65–82.11)
Inflammatory/healthy	0.56	0.40–0.73	>0.05	>0.56	62.50 (35.43–84.80)	69.84 (56.98–80.77)
Neoplastic/healthy	0.94	0.89–0.99	<0.0001	>0.69	89.47 (66.86–98.70)	84.13 (72.74–92.12)

AUC: area under receiver operating characteristic curve, CI: confidence interval, Se: sensitivity, Sp: specificity.

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
