# Peer review of "Sonographic Evaluation of Medial Iliac Lymph Nodes-to-Aorta Ratio in Dogs"

_vetsci, 2020, doi:10.3390/vetsci7010022_

Round 1
Reviewer 1 Report
english language should be strongly reviewed
Author Response
dear reviewer, thank you for your comments.
as you suggested, the text has been revised for the English language by a native English speaking colleague
simonetta citi
Reviewer 2 Report
Comments to the author´Sonographic Evaluation of Medial Iliac Lymph Nodes-to-Aorta Ratio in Dogs`.
The paper investigates the size of the medial iliac lymph nodes compared to the aorta diameter and also the difference in size between inflammatory and neoplastic lymph nodes in dogs.
Nice design and interesting idea.
Line 11: Considering that there is not a standard measurement:Rephrase. There are no reference values for size in healthy dogs.
Line 36: are anatomically located ventral to…
Line 37: and lymphadenomegaly can be radiographically diagnosed.
Line 42: which may limit their visualization: Delete
Line 68-69: If the reported sizes correlate to the body weight, then they are not necessarily correlated with type of breeds. Please report the size in the body weight groups. Do they refer to the medial iliac lymph nodes? Please specify.
Line 73: The ultimate objective was to find a reference range in healthy dogs normalized by the size. A second objective was to investigate the change in size of the medial iliac lymph nodes in neoplastic and inflammatory diseases.
Line 87: Which included examination of….
Line 102: What does good temperature mean? please specify. Two operators gently manually restrain the dogs if…
Lien 103: Recumbency- also in the following sentences.
Line 113: Length, defined as maximal craniocaudal extension, (L) was measured… Please similarly define thickness and height.
Line 118-119: Why was the diameter of the Aorta measured in transverse and longitudinal scan? I guess to assess variability? Were the measurements taken only once (once longitudinal and one transverse scan), were they averaged?
Line 140: Please state the body weight mean.
Line 141: Please state the breeds included. With this number of subjects, you can use mean and standard deviation of the age and weights to describe the distribution.
Line 144: Please state the breeds included. With this number of subjects, you can use mean and standard deviation of the age and weights to describe the distribution.
Line 249: Could you please comment on the results about the length and thickness of the medial iliac lymph nodes? You write that there are statistically significant differences between left and right. Of which entity? Were they already reported?
Line 156: Only altered lymph nodes
Line 177: Once explained that the aorta diameter refers to the longitudinal diameter, you don`t have to specify it every time.
Line 186: I think the manuscript will profit for a short additional information, that is how and if the ratio between lymph nodes size and aorta diameter changes in body weight classes. We assume that the ratio would be similar independently by the body weight but it has to been proven. Similar to the cited Mayer`s article, I would describe and stress if this `normalized measurement` would than be free of body weight` influence.
Figure one: If I understand the data correctly, there is no statistical significant difference in the height of the right versus left medial iliac lymph nodes. In this case, I will group the graphs with the mean of the left and right lymph nodes and display then the mean of the height divided by the aortic diameter in healthy and diseased animals; in healthy and neoplastic patients; and healthy and inflammatory patients.
Line 219-221: This sentence can be proved only reporting the values of the ratio in different body weight classes as suggested. Otherwise it is only a hypothesis.
Line 253-4: Please rephrase
Line 239: It is not possible
Line 260: Other 2 important limitations are: The normal group was obviously not sampled, and, even if unlikely, we can not completely exclude disease at microscopic level. The diagnosis of neoplastic and inflammatory disease also relayed on cytologic examination, which can also lead to misdiagnosis.
Line 262: I would rephrase. This study does not investigate which is the first sign of diseased lymph nodes, it could happen that the size is still normal and the echogenicity varies.
Line 267: Please rephrase. You assume that the change in size is always the first sign of pathology but that is not always true. Or please provide references.
Round 2
Reviewer 2 Report
Definitely improved.
Line 20-21: Be consistent with punctuation/decimal
Line 149: Statistically, you can describe data distribution with mean and standard deviation OR with median and range. Change it accordingly.
Line 199: Spearman’s rank correlation coefficients and linear regression models didn't showed a correlation between body weight of dogs and height of both the right and left medial iliac lymph nodes (p > 0.05).Is there a correlation between Aorta Diameter and body weight?
That is in contradiction with line 254-255?
Line 254-257: Still to justify this sentence, you have to report the ratio (lymph node height/aorta diameter) in different weight classes and show that this ratio does not change.
Line 224: ratio of the meanof right and left lymph nodes height (please specify in A, B and C).
Author Response
Answers to Reviewer 2 (round 2). The second change are red and bold
Line 20-21: Be consistent with punctuation/decimal
Response: As suggested by the reviewer, the number has been changed
Line 149: Statistically, you can describe data distribution with mean and standard deviation OR with median and range. Change it accordingly.
Response: As suggested by the reviewer, the sentence has been changed
Line 199: Spearman’s rank correlation coefficients and linear regression models didn't showed a correlation between body weight of dogs and height of both the right and left medial iliac lymph nodes (p > 0.05). Is there a correlation between Aorta Diameter and body weight?
Response: We sorry to the reviewer, but we have written incorrectly: there is a correlation between aorta and heigth of lymph node (lines 194-196) and between aorta and body weight (line 255). There isn’t correlation between body weight of dogs and ratio between height and aorta’s diameter (lines 198-200)
That is in contradiction with line 254-255?
Response: The sentence has been changed accordling with results
Line 254-257: Still to justify this sentence, you have to report the ratio (lymph node height/aorta diameter) in different weight classes and show that this ratio does not change.
Response: this consideration is based on a mistake reported in the previous draft, which has been corrected; therefore we do not consider it necessary to include the weight classes
Line 224: ratio of the meanof right and left lymph nodes height (please specify in A, B and C).
Response: As suggested by the reviewer, the sentence has been changed